# HARMON: Whole-Body Motion Generation of Humanoid Robots from Language Descriptions

**Zhenyu Jiang[1,2*]** **Yuqi Xie[1,2*]** **Jinhan Li[1†]** **Ye Yuan[2]** **Yifeng Zhu[1]** **Yuke Zhu[1,2]**

[1]The University of Texas at Austin    [2]NVIDIA Research

{zhenyu, yukez}@cs.utexas.edu

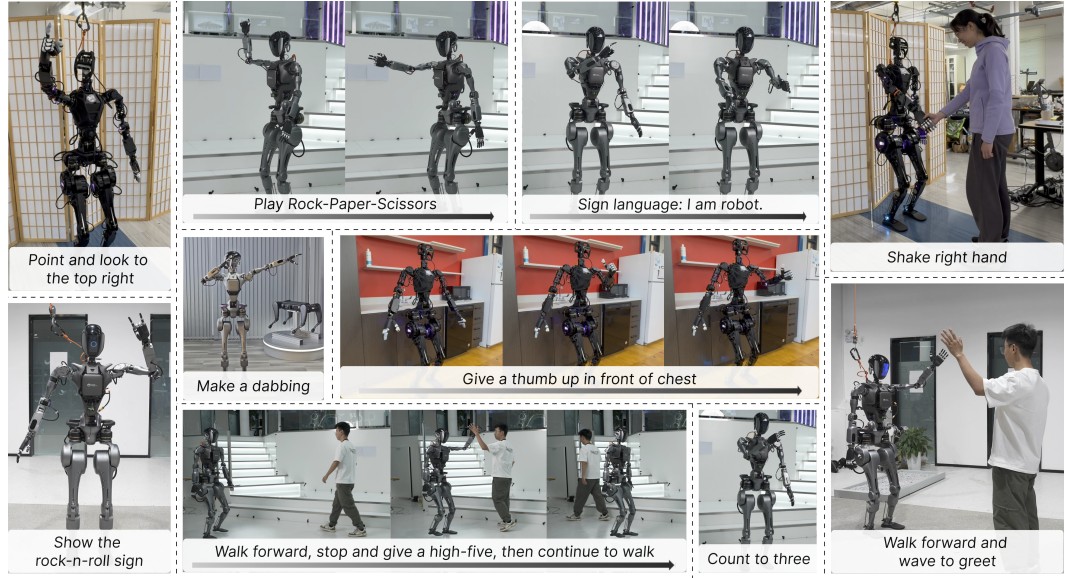

Figure 1: We generate diverse whole-body humanoid motions from free-form language descriptions and execute these motions on the real humanoid robot.

**Abstract:** Humanoid robots, with their human-like embodiment, have the potential to integrate seamlessly into human environments. Critical to their coexistence and cooperation with humans is the ability to understand natural language communications and exhibit human-like behaviors. This work focuses on generating diverse whole-body motions for humanoid robots from language descriptions. We leverage human motion priors from extensive human motion datasets to initialize humanoid motions and employ the commonsense reasoning capabilities of Vision Language Models (VLMs) to edit and refine these motions. Our approach demonstrates the capability to produce natural, expressive, and text-aligned humanoid motions, validated through both simulated and real-world experiments. More videos can be found on our website https://ut-austin-rpl.github.io/Harmon/.

**Keywords:** Humanoid Robot, Whole-Body Motion Generation

## 1 Introduction

Humanoid robots have great potential for seamlessly integrating into the human world due to their human-like physique. We envision these robots operating in human-centered environments and coexisting with people in shared physical spaces. To deploy humanoid robots to work and live with

---

[†] This work was done while Jinhan Li was a visiting researcher at UT Austin.

[*] Equal contribution

8th Conference on Robot Learning (CoRL 2024), Munich, Germany.

us, the robots should possess the capability to understand natural language instructions akin to our everyday conversations. Furthermore, the ability to exhibit human-like behaviors will render these robots more social and predictable, thus enhancing effective and safe collaborations between humans and robots. To this end, this work focuses on developing a method for generating diverse behaviors of humanoid robots from text descriptions. Our approach takes a natural language description of the desired motions as input and produces corresponding whole-body motions.

Training a model to generate natural humanoid motions from language descriptions is challenging due to the lack of large-scale datasets of language-motion pairs. Fortunately, the narrow embodiment gaps between humans and humanoid robots unlock vast amounts of human-centered data sources for model training. Particularly, there is an abundance of large-scale human motion data [1–5]. These data, paired with language descriptions, capture a wide range of human motions. These human motions can be mapped to humanoid robots as informed priors for motion generation. Nonetheless, discrepancies between the human model and the humanoid hinder the effectiveness of direct motion retargeting. First, head and finger motions are typically absent from the human motion dataset, which consequently cannot produce humanoid motions for these body parts. Yet, these motions are critical for the expressiveness of the robot's whole-body motion. Further, retargeting cannot precisely replicate human motion on humanoids due to their kinematic constraints, potentially changing the semantic meaning and legibility of the motion.

In this work, we introduce HARMON (**H**um**a**noid **R**obot **Mo**tion Ge**n**eration), which generates whole-body humanoid motions from free-form language descriptions by incorporating human motion priors. We employ PhysDiff [6], a diffusion-based generative model trained on large-scale human motion datasets to generate human motions from language descriptions. PhysDiff incorporates physics constraints during the diffusion process and generates physically plausible human motions. It generates human motion as a sequence of SMPL [7] parameters. We then retarget the human motion to a simulated humanoid robot using inverse kinematics. As mentioned earlier, the generated motion from PhysDiff does not involve hand and finger motions, and retargeting errors can lead to the misalignment between the motion and the language description. To address these issues, we leverage the commonsense reasoning capability of Vision Language Models (VLMs) to edit the humanoid motion. Given a rendering of the humanoid motion and its language description, the VLM generates head and finger motions and refines the arm motion. To realize the whole-body motion on the real robot, we separate the upper- and lower-body motions and control locomotion and upper-body motion independently. Benefiting from the human motion prior and VLM-based motion editing, we can generate natural humanoid motions that align with the language description and execute the motions on simulated and real humanoid robots.

We use a Fourier GR1 humanoid robot for the simulation and real-world experiments. We curate a test set with texts from the HumanML3D test set and LLM-generated motion descriptions and conduct a human study to evaluate the quality of the generated motion. HARMON demonstrates natural and text-aligned humanoid motions and is favored by human evaluators on 86.7% of test cases. Furthermore, we execute the generated motions on the physical robot and illustrate diverse and expressive humanoid motions in the real world.

## 2 Related Work

**Humanoid Motion Control.** Humanoid motion control has been widely studied in the computer graphics community, where human avatars are simulated to demonstrate diverse and physically realistic behaviors [8–24]. These methods can demonstrate physically realistic motions on the humanoid avatar by imitating human motions with reinforcement learning. However, deploying them on the real robot is difficult since they use an unrealistic humanoid model (SMPL robot with 23 ball joints and no torque limit). Recent studies [25–27] have explored the imitation of human motions on real and simulated humanoid robots. These approaches involve retargeting human motion to humanoid robots and using reinforcement learning (RL) to train the robots in simulation. Some of these studies [26, 27] have successfully deployed the trained policies to real-world robots. Our work focuses on text-conditional humanoid motion generation. Yoshida et al. [28] also address

this problem by directly generating humanoid motion using a Large Language Model. However, their generated motions often appear unnatural and rigid. In comparison, we incorporate human motion priors to produce more natural humanoid motions.

**Human Motion Generation.** Recently, significant developments have occurred in human motion generation. Early studies [29–36] generated human motion deterministically. Subsequently, stochastic generative models were applied to human motion generation, employing GANs [37] or VAEs [38–41] to create human motions from various conditions, such as action labels and texts. More recently, diffusion models [42, 6, 43] have also been utilized for human motion generation. These models can accept various conditions such as text or keyframes and produce diverse and natural human motions. Additionally, some recent works [44, 45] tokenize human motions and use transformer-based autoregressive models to generate human motion through next-token prediction.

In our work, we use human motion generated from text as an initialization for humanoid motion. While these text-conditioned human motion generation models provide a good prior for humanoid motion, the structural differences between the human models in these frameworks and the real humanoid necessitate further refinement of the humanoid motion.

**Foundation Models for Robot Control.** Foundation models, such as Large Language Models (LLMs) and Vision-Language Models (VLMs), have shown exceptional performance as high-level semantic planners for tasks involving embodied agents and robotics [46–55]. Recent studies [56–58] have started to explore their potential for learning low-level robot behaviors. For example, L2R [58] uses few-shot examples to prompt LLMs to generate reward functions for robot training, while Eureka [56] eliminates the need for few-shot examples by employing an evolutionary search system to iteratively propose improved reward functions. RL-VLM-F [59] generates reward functions for agents to learn new tasks with feedback from VLMs. We leverage the zero-shot commonsense reasoning capabilities of VLMs to evaluate and edit humanoid motions, improving the alignment between humanoid motions and corresponding language descriptions.

## 3 Method

We study the problem of generating humanoid motions from language descriptions. From a text description $X$, we aim to generate a sequence of robot joint configurations $Q = \{\mathbf{q}_1, \cdots \mathbf{q}_T\}, \mathbf{q}_i \in \mathbb{R}^c$. Here, $T$ is the sequence length, and $c$ is the number of joints of the humanoid robot.

Fig. 2 depicts our proposed method, HARMON. Firstly, we generate human motion based on the language description and retarget this human motion to create the initial humanoid motion (Sec. 3.1). To improve the alignment between the humanoid motion and the language description, we employ a VLM to generate finger and head motions and iteratively adjust the body motion (Sec. 3.2). Finally, we execute the generated motions on the real humanoid robot (Sec. 3.3).

### 3.1 Retargeting Text-Conditioned Human Motion

Training a model to generate humanoid motions from language descriptions directly is challenging due to the absence of a paired dataset. We utilize a human motion generation model to generate human motions from language descriptions, which we then retarget to the humanoid robot.

Given the text description $X$, we use PhysDiff [6], a physics-guided motion diffusion model, to generate the corresponding human motion. The output is a sequence of SMPL [7] parameters $P = \{(\theta_1, \mathbf{t}_1), \cdots, (\theta_T, \mathbf{t}_T)\}$, where $\theta_i$ is the joint rotations and $\mathbf{t}_i$ is the root translation at time step $i$. The SMPL model also includes a body shape parameter $\beta \in \mathbb{R}^{10}$, encoding attributes such as height and size. Given $\theta_i, \mathbf{t}_i, \beta$, the positions of each human joint $J_i = \mathcal{S}(\theta_i, \mathbf{t}_i, \beta)$ are computed using the SMPL model $\mathcal{S}$. Each $J_i \in \mathbb{R}^{24 \times 3}$ contains the positions of 24 human joints at time step $i$.

Before retargeting human motion to humanoids, we first minimize the disparity between human body shape and humanoids to ensure the robots can reach human joint positions. Inspired by He et al. [27], we set the SMPL model and humanoid model to the same T pose, select 17 corresponding joint pairs on both models, and minimize the joint position differences. We employ the Adam optimizer [60] to minimize the joint position difference by optimizing $\beta$. The optimized $\beta^*$ is subsequently used to compute joint positions from the SMPL parameters.

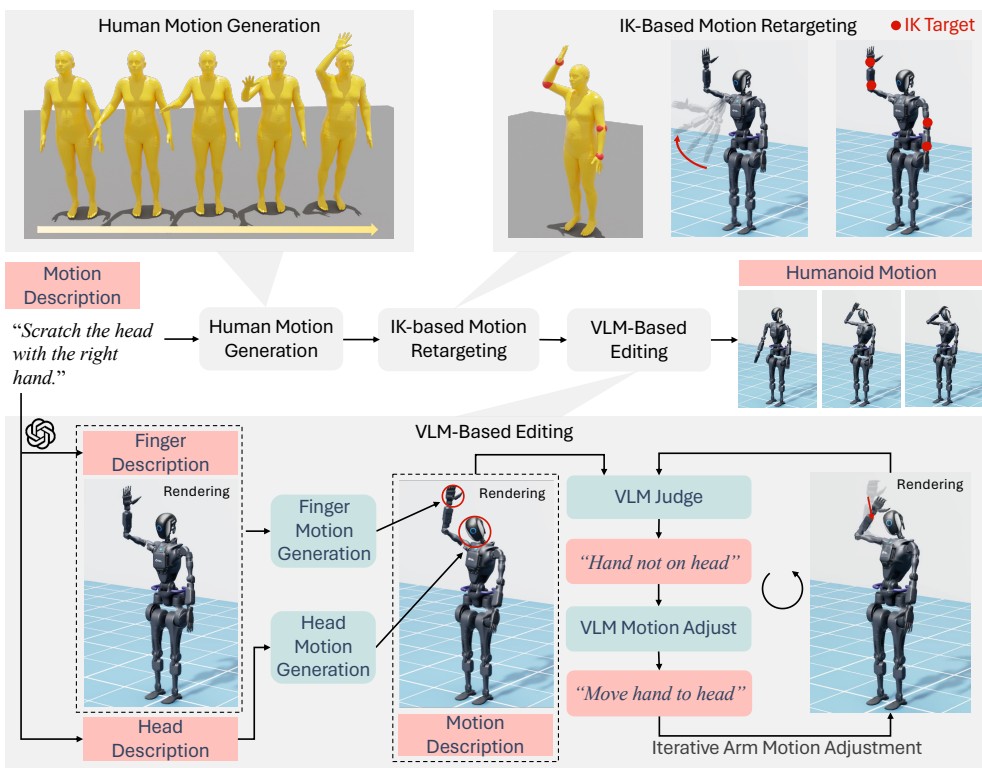

Figure 2: **Overview of HARMON**. Given the language description of a motion, we first generate corresponding human motion and retarget it to the humanoid using inverse kinematics. Next, we utilize a VLM to refine the humanoid motion. This process involves extracting finger and head motion descriptions from the initial language description and generating the corresponding motions using the VLM. Given the rendered humanoid motion, the VLM iteratively evaluates and adjusts the motion to ensure alignment with the language description. Finally, HARMON generates whole-body humanoid motion that accurately aligns with the language description.

Next, we utilize the inverse kinematics (IK) solver from pink [61] to align the key joints of the humanoid with those of the human model by optimizing the humanoid's joint configuration $\mathbf{q}$. The key joints include the wrists, elbows, shoulders, knees, and ankles. Given the current joint configuration of the humanoid and the target positions of the key joints, the solver calculates the joint velocities to drive the key joints to their target positions. We sequentially set the target as the spatial position of the SMPL model and update the joint configuration with the results from the IK solver at each time step. After iterating through the SMPL parameters for each time step, we obtain a sequence of robot joint configurations, $Q_r$, from the retargeting process. Controlling the robot to follow this sequence results in humanoid motion that closely mimics the generated human motion.

## 3.2 VLM-Based Humanoid Motion Editing

The retargeting process initializes humanoid motion based on the generated human motion. However, due to the differences in the kinematic structures between the SMPL human model and the humanoid, the retargeted motion might not fully align with the intended language description. Specifically, the humanoid can actuate the neck and dexterous hands, whereas the generated human motion lacks head and finger movements. Consequently, the retargeted motion does not fully exploit the humanoid's potential for expressive motion. Additionally, the retargeting process cannot precisely replicate human motion on the humanoid, potentially resulting in a humanoid motion with a different semantic meaning from the original human motion. These factors can cause a misalignment between the humanoid motion and the language description. To address this issue, we render the retargeted motion into videos and utilize GPT-4 to edit the humanoid motion for better alignment.

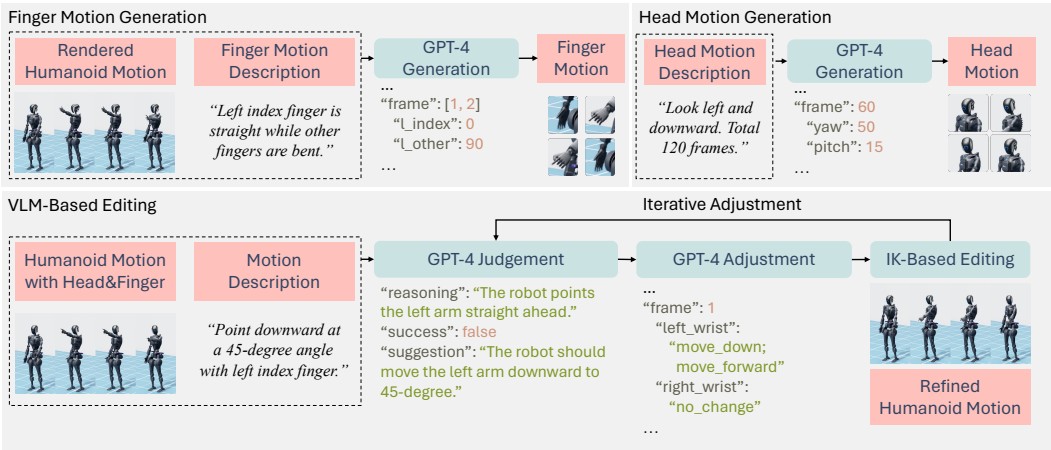

Figure 3: **VLM-based motion editing**. **Top left**: GPT-4 generates finger motions at keyframes based on the rendered humanoid motions and the finger motion description. **Top right**: GPT-4 identifies keyframes and generates head motions from the head motion description. **Bottom**: GPT-4 iteratively adjusts arm motion by evaluating and refining the rendered humanoid frames based on the motion description.

**Finger and Head Motion Generation.** To maximize the expressiveness of the humanoid motion, it is essential to generate finger and head movements and integrate them with the whole-body motion from the retargeting process. As illustrated in Fig. 3, the finger and head motions are generated separately, each with its own motion description. We employ GPT-4 to extract these specific descriptions from the original motion description. This approach allows the VLM/LLM to concentrate on generating precise finger and head movements while avoiding unnecessary motion generation when these parts are not involved.

The finger motion needs to be coordinated with the arm motion. To achieve this, we use GPT-4 to observe the rendered video of the retargeted whole-body motion and generate the corresponding finger movements. Since GPT-4 only accepts images as input, we sample four frames at equal intervals from the video. Although more frames might capture additional details, we found empirically that the existing VLMs exhibit reduced reasoning ability when input sequences are too long. For our task, four frames are sufficient for the VLM to generate the finger motion accurately. We input these four frames along with the finger motion description into GPT-4, prompting it to generate hand joint configurations $\mathbf{q}_i^f \in \mathbb{R}^{n_f}$ for each interval, where $n_f = 12$ represents the total number of finger joints. The results for different intervals are then concatenated into a sequence of finger joint configurations, $Q_f$.

The head motion is more independent and has lower dimensionality than the finger motion. Therefore, we use an LLM, specifically GPT-4, to directly generate head movements from the head motion description. We provide GPT-4 with the head motion description, the total number of frames, and the frames per second (FPS) as input. GPT-4 then determines the joint configurations $\mathbf{q}_i^h \in \mathbb{R}^3$ for the three neck joints at keyframes. By interpolating between these keyframes, we obtain smooth head movements, represented by the sequence of joint configurations $Q_h$. The key frame indices are determined autonomously by GPT-4, allowing for the generation of high-frequency head motions.

**Iterative Motion Adjustment.** The retargeted motion may not align with the language description for two primary reasons: 1) the generated human motion may not accurately reflect the language description, and 2) the retargeting process may alter the human motion, resulting in humanoid motion with a different semantic meaning. To address these discrepancies, we implement an iterative motion adjustment schema to align the humanoid motion with the language description.

As shown in the bottom row of Fig. 3, we employ a judgment agent and an adjustment agent to ensure alignment. Similar to the process for generating finger motion, we select four frames from the generated motion video at equal intervals. These frames, along with the human motion description, are provided to the judgment agent. GPT-4V first generates a caption describing the actions of the

humanoid in the video. Then, it assesses whether the actions match the motion description and provides suggestions for improvement. In the next step, the same four screenshots and the generated suggestions are input into the adjustment agent. The adjustment agent then predicts the necessary adjustments to align the motion with the provided suggestions.

The key to effective motion edits is providing an intuitive interface for the adjustment agent. Directly allowing the VLM to edit the joint configurations of retargeted motion $Q_r$ is challenging due to the non-intuitive mapping between joint configurations and motion. Additionally, editing lower body motion is not particularly meaningful for this project, as the lower body is controlled by a separate controller in the real robot experiment. Therefore, our focus is on the upper body motion, which demonstrates richer semantics and is easily controllable on the real robot. We design a set of control primitives that move the left and right wrists in specific directions using inverse kinematics, such as moving up/down or towards the head/chest. Details about these primitives are provided in the appendix. The adjustment agent can combine these primitives to create motion adjustments. We then apply these adjustments, render a new video, and return to the judgment agent to start a new evaluation round. This process is repeated until the judgment agent confirms that the motion aligns with the language description or the adjustment process exceeds two rounds. Since our control primitives are limited to spatial wrist movements, we include an additional step in the judgment agent to determine if the current motion can be improved by the editing process based on the language description. If improvement is unlikely, the process is skipped.

If any adjustments are made, we use the body joint configuration sequence $Q_b$ resulting from the final round of editing. If no edits are necessary, we directly use the retargeted motion $Q_r$ as the final body joint configuration sequence $Q_b$. Finally, we combine $Q_b$, $Q_f$, and $Q_h$ to form the complete body joint configuration sequence $Q^*$.

### 3.3 Motion Execution on the Real Robot

Directly executing the whole body joint configuration sequence $Q^*$ on the real humanoid is infeasible because the kinematic motion does not account for the robot's dynamics and balance. Therefore, following Cheng et al. [26], we separate the lower- and upper-body motion in the real robot experiment. We simplify the lower body motion into locomotion commands and utilize a Zero Moment Point (ZMP)-based [62] controller for locomotion. These locomotion commands are extracted from the trajectory of the humanoid robot's pelvis projected onto the ground plane. Simultaneously, we execute the upper-body motion from the generated joint configuration sequence $Q^*$ on the real robot using joint position control. By separating locomotion and upper-body motion, we can successfully execute our generated motion on the real robot.

## 4 Experiments

### 4.1 Evaluation Setup and Baselines

Evaluating whole-body humanoid motion generation is challenging due to the absence of datasets containing paired language and humanoid motion data. We conduct a human study to assess our generated motions. We curate a test set of approximately 50 language descriptions of motions. The first part of this test set comprises texts randomly sampled from the HumanML3D [1] test set, focusing primarily on body motions without involving head and finger movements. To evaluate whole-body humanoid motions, we use GPT-4 to generate descriptions that include head and finger motions, forming the second part of the test set.

We generate humanoid motions from these descriptions using HARMON and compare them against three baselines:

**VLM-based Motion Generation**. Inspired by Yoshida et al. [28], this baseline generates humanoid motions directly using a Large Language Model (LLM). In this approach, we exclude the human motion prior from HARMON and initiate VLM-based motion editing from a static SMPL T-pose. Since VLM-based editing only modifies upper-body joints, we incorporate the lower-body motion from HARMON for a fair comparison. This allows us to evaluate the significance of the human motion priors.

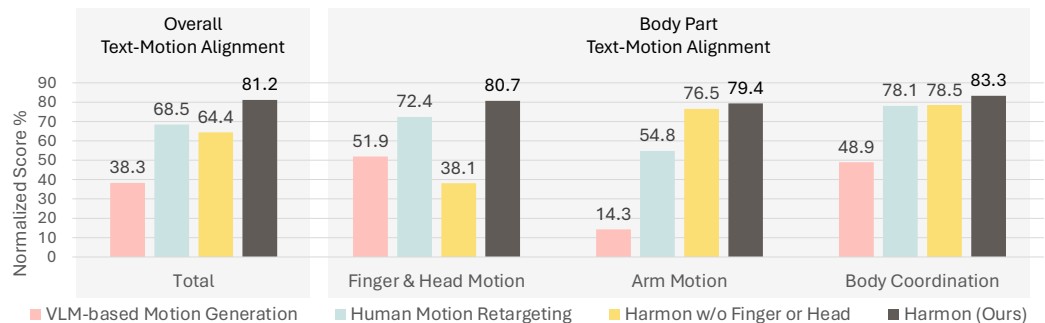

Figure 4: **Quantitative results of human study**. A higher normalized score indicates a better alignment between the humanoid motion and the language description.

**Human Motion Retargeting**. This baseline uses the retargeted human motion directly as the body motion. To ensure a fair comparison with HARMON, we add the finger and head motions from HARMON to create the complete whole-body motion. This comparison helps us assess the effectiveness of the iterative motion adjustment.

**HARMON w/o Head or Finger**. This is an ablated version of HARMON where the head and finger joints are defaulted to zero position rather than generated.

By comparing HARMON with these baselines, we aim to evaluate the importance of human motion priors, iterative motion adjustment, and the head and finger motion generation.

The resulting humanoid motion from HARMON and the baseline models are rendered into simulation videos and shown to participants in a human study. For each video, participants evaluate~~evaluate~~ whether the humanoid's movements align with the text, focusing on three specific aspects: 1) finger and head movements, 2) arm movements, and 3) overall body coordination. The movement of these body parts reflects the expressiveness, accuracy, and naturalness of the generated motion. In total, we collect assessments for 1728 results from different methods provided by 12 participants.

## 4.2 Evaluation of Humanoid Motion Generation

We first evaluate the overall alignment between the generated humanoid motion and the corresponding language description for each method. Each aspect (finger and head movements, arm movements, and overall body coordination) is scored by participants, contributing one point if marked as correct. We aggregate the scores across all aspects, results, and participants to obtain a total score for each method. The total scores are then divided by the maximum possible score to calculate the normalized scores. The overall normalized scores are displayed on the left part of Fig. 4. HARMON achieves a high normalized score of 81.2%, significantly outperforming the baselines. The VLM-based motion generation baseline receives the lowest normalized score, underscoring the importance of incorporating human motion priors in humanoid motion generation.

We conduct a more detailed analysis of the text-motion alignment of different body parts for each method. We compute the scores for each body part separately and visualize the results on the right part of Fig. 4. The VLM-based motion generation baseline consistently scores the lowest across almost all body parts. This result highlights the difficulty for VLM in generating accurate motions directly from language descriptions without a reasonable initialization from human motion priors, especially for complex motions. The human motion retargeting baseline shows a lower normalized score for arm motions, demonstrating that the VLM-based iterative motion adjustment improves alignment between arm motions and the language description. The baseline without finger or head motions naturally scores lower in these aspects. However, the normalized score is non-zero because some test motion descriptions do not involve finger motions.

Additionally, we visualized some of HARMON's results in Fig. 5. The retargeted human motion provides a solid initialization for humanoid motion, especially for high-frequency actions like clapping (first row). These visualizations also highlight various instances where the retargeted human motion

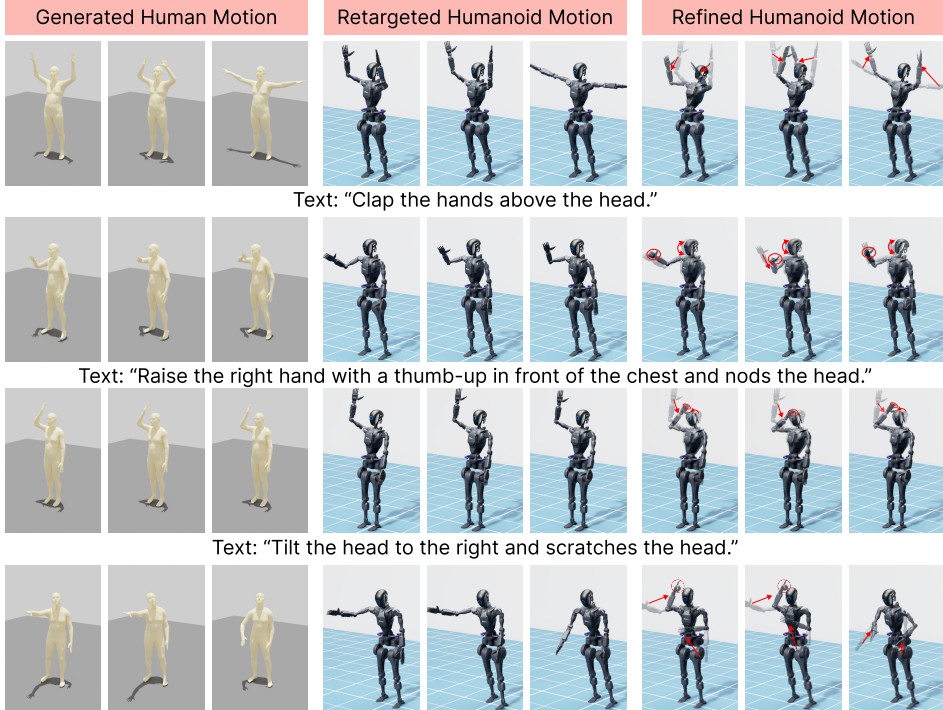

| Generated Human Motion | Retargeted Humanoid Motion | Refined Humanoid Motion |
|---|---|---|

Text: "Clap the hands above the head."

Text: "Raise the right hand with a thumb-up in front of the chest and nods the head."

Text: "Tilt the head to the right and scratches the head."

Text: "Point diagonally upwards to the left with index finger and places the left hand on chest."

Figure 5: **Qualitative results of HARMON**. We highlight the generated head and finger motions with red circles and the motion adjustment with red arrows.

misaligns with the language description and how VLM-based motion editing corrects these issues. For example, in the second row, VLM-generated finger and head motions significantly enhance the expressiveness of the humanoid motion. In the third row, the retargeting error causes the humanoid motion to deviate from the original human motion, but the iterative motion adjustment successfully realigns it with the text description. Additionally, there are cases where the generated human motion itself did not align well with the language description, as seen in the fourth row. Here, iterative motion editing also improved the alignment, demonstrating its effectiveness in refining the motion to match the language description better.

### 4.3 Real-Robot Deployment

When executing the generated motions on the real robot, we decouple locomotion from upper-body movements to maintain the robot's balance while demonstrating the generated motions. We tested both standing motions and motions requiring locomotion on real GR1 humanoids. We illustrate some example results in Fig. 1. Videos showcasing the real robot executing the generated motions are on the project website.

## 5 Conclusion

This work addresses the problem of generating humanoid motion from free-form language descriptions. We introduce HARMON, which utilizes human motion priors and the commonsense reasoning capabilities of VLMs for humanoid motion generation. Our results demonstrate that HARMON can produce natural, expressive, and text-aligned humanoid motions, which are executable on real humanoid robots. Our iterative motion adjustment process relies on a fixed set of intuitive control primitives. These primitives struggle when the generated human motion significantly deviates from the language description, especially for high-frequency motions. A potential future direction is to enable the VLM to generate its free-form control primitives. In this work, we separate lower- and upper-body control for the real robot. As a result, some generated motions may cause the robot to lose balance or collide with itself. We aim to incorporate RL-based whole-body control for more robust execution on the real robot in future work.

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
