# OpenReview forum: "Harmon: Whole-Body Motion Generation of Humanoid Robots from Language Descriptions"
_robot-learning.org/CoRL/2024/Conference — CoRL 2024_

### Official Review · Reviewer_Ynst · 2024-07-17
**Comments from reviewer Ynst**

**Originality:** 4
**Technical Quality:** 4
**Clarity Of Presentation:** 4
**Potential Impact:** 3
**Recommendation:** 3
**Confidence:** 4

**Review:**

Strengths:
- The investigated topic is interesting and well-motivated.
- The paper is well-written and easy to read.
- The proposed method, by leveraging human motion prior and VLM-based motion refinement, shows good performance in both simulation and real-world humanoid robots, especially when finger and head motion is needed.
- Although leveraging human motion datasets has been discussed in previous papers, it is good to see with VLM-based refinement, an IK-based method can already achieve good performance in real-world robotics.

Weaknesses:
- I don't have major concerns for the paper. The reason I give weak accept instead of strong accept is that the method is based on IK, which usually gives physically infeasible motions. Although the paper uses VLM-based refinement together with separated lower- and upper-body control for the real robot, the resulting motion can still be improved a lot. For example, the walking gait in the demo looks not very robust. This is also discussed in the limitation section.
- It will be good to include some failure cases.
- In Lines 48-49, what do you mean by "we decouple the upper- and lower-body motions and control locomotion and upper-body motion separately"?

**Quality Of The Limitations Section:**

3

**Questions For Rebuttal:**

See the weaknesses above.

**Robotics Focus:**

4

**Summary Of Paper:**

An IK-based humanoid motion generation approach with motion prior extracted from existing human motion datasets following a VLM-based motion refinement step.

**Summary Of Recommendation:**

The paper proposes a pipeline to generate language conditioned motions for humanoid robots. The method extracts human motions prior from existing datasets and refines the motion via VLM for better retargeting. However, the adopted IK solver may cause improper motions and can be improved. Therefore, my recommendation is weak accept.

---

### Official Review · Reviewer_HfRF · 2024-07-20
**Interesting use of VLMs but needs additional analysis**

**Originality:** 3
**Technical Quality:** 3
**Clarity Of Presentation:** 4
**Potential Impact:** 3
**Recommendation:** 3
**Confidence:** 4

**Review:**

The idea of using a VLM to fine-tune the retargeted motion is novel and creates a fully automatic process to generate humanoid motion from text instructions. The authors deploy the trajectories on a real robot to further provide some convincing evidence of their work. The paper is well-written and easy to understand.

However, there as some concerns:
1. Since the entire approach relies on the VLM being able to comprehend the images being fed into it (possibly out-of-distribution for ChatGPT), I believe the paper can benefit from some additional exploration about best practices to adopt here.
For example:
a) does the image need to be a photo-realistic rendering of the robot? Or can it just be collision shapes?

b) Does the approach generalize to different robot head and hand designs (visual differences)?

c) what happens when we increase the number of frames. How much does the performance deteriorate?

d) Is the approach tied up with the specific VLM model? What happens when you switch the VLM?

2. From my understanding, the approach cannot handle any differences emerging from the dynamics of the two morphologies. It is not aware of any torque or velocity limits when generating the trajectories. It is essentially just stepping through a list of poses. While this can be further used to train an imitation learning based RL policy, the current version of the paper does not tackle this.

3) The control primitives limit the extent of modifications the VLM can perform on the trajectory. The VLM can only select from a predefined set of primitives which are hard coded to perform very specific actions. For example, there is a primitive called move_up: which moved the target up by 20 cm. The approach can benefit from using a parameterized set of control primitives instead of hard-coded movements.

**Quality Of The Limitations Section:**

2

**Questions For Rebuttal:**

1. What is the difference between the “human motion Retargeting” and Harmon w/o Finger or Head” baselines?
2. Can the framework support more fine-grained modifications that are not supported by the current control primitives?
3. Follow up to the above question, can the VLM reason about parameterized control primitives? eg: move target by x, where x has to be generated by VLM.
4. How are self-collisions handled by the approach?

**Robotics Focus:**

4

**Summary Of Paper:**

This paper presents an approach to create humanoid motion trajectories from text instruction. The approach retargets upper-body human motion trajectories to a humanoid robot using IK and then uses a fine-tuning step that leverages the common-sense reasoning capabilities of VLMs. The paper evaluates the approach using a human study and showcases hardware results by executing the trajectory on a humanoid robot.

**Summary Of Recommendation:**

The paper has an interesting idea but additional analysis needed to make it useful for the robotics community

---

### Official Review · Reviewer_4CM1 · 2024-07-22
**A simple approach to an interesting problem with good hardware results**

**Originality:** 3
**Technical Quality:** 4
**Clarity Of Presentation:** 4
**Potential Impact:** 3
**Recommendation:** 3
**Confidence:** 4

**Review:**

- The paper extends the use of vision-language models to improve the head and fingers in retargetted human motion starting purely from text.
- The paper is very clearly written and easy to follow. The figures are also very informative and aid the understanding of the paper.
- The paper will likely have a reasonable impact for attempting a VLM in the loop for motion generation and refinement for hands and head motion, however, the exact method proposed here will severely limit the applicability of the exact algorithm proposed in this method because of a key limitation -- the use of predefined primitives to allow the VLM to easily update the retargetted human motion.
- The paper ablates a majority of the aspects of the proposed method and also compares with the relevant baselines showing superior performance. There is one aspect of the proposed method whose empirical importance is not very clear (see below).
- The limitation section addresses the key limitations of the proposed method on point. The use of predefined primitives limit the scope of VLM in generating motions.

**Quality Of The Limitations Section:**

3

**Questions For Rebuttal:**

- For "VLM based motion generation" baseline, it might be a fairer comparison to compare upper body only since attaching a lower body which is incoherent might make it a lower quality output overall.
- If I understand correctly, HARMON w/o head and finger uses the default head and finger values from the previous retargetted prediction?
- It looks like another baseline would be "VLM-based Motion Generation" + starting from a motion prior model + iterative refinement (without explicit hand and head module) but still using the additional text description from GPT4. Essentially, this will highlight the importance of the hand and head motion generation module before iterative refinement.

**Robotics Focus:**

4

**Summary Of Paper:**

The paper proposes a method called HARMON. The whole system starts with a language description, uses that to generate a human mesh which is retargetted using an IK based solver, and then uses a VLM to generate and iterate head and finger motions. The proposed method is shown on hardware and compares against various ablations of the proposed method.

**Summary Of Recommendation:**

The paper takes an interesting approach to handling head and finger refinement of generated motions, presents compelling real world results, but is lacking some comparisons which help understand the importance of the proposed method.

---

### Decision · Program_Chairs · 2024-09-04

**Decision:**

Accept

**Comment:**

Evaluating this paper is very challenging as there is very little to no prior art on the topic. The reviewers are overall mildly positive but not enthusiastic about the paper. The reviewers raise the concerns that the approach is very limited due to the required primitives which are very vague, e.g. move hand in direction x. Furthermore, the approach relies on very elaborate prompts and it is unclear how sensitive the system is to small changes in the prompts. It is unclear how much the VLM really understands the task or just understands how to match words in the prompt.

As it is the first CoRL where such work is appearing it is hard to evaluate. On the positive side, the authors provide the full prompts in the appendix, apply their method to a humanoid and present their approach clearly. On the negative side, the performed experiments are very simple and the robot movement only vaguely resembles the described simple movement aka wave hand.

I would recommend accepting the paper as VLM + robotics is a hot topic and the authors did one approach and clearly described their approach. However, I would not defend my recommendation.